# Process Design of Continuous Powder Blending Using Residence Time Distribution and Feeding Models

**DOI:** 10.3390/pharmaceutics12111119

**Published:** 2020-11-20

**Authors:** Martin Gyürkés, Lajos Madarász, Ákos Köte, András Domokos, Dániel Mészáros, Áron Kristóf Beke, Brigitta Nagy, György Marosi, Hajnalka Pataki, Zsombor Kristóf Nagy, Attila Farkas

**Affiliations:** Department of Organic Chemistry and Technology, Budapest University of Technology and Economics (BME), Műegyetem rakpart 3, H-1111 Budapest, Hungary; gyurkes.martin@mail.bme.hu (M.G.); madlajos96@gmail.com (L.M.); kote2ster@gmail.com (Á.K.); domokos.andras@mail.bme.hu (A.D.); meszarosdani06@gmail.com (D.M.); bekearon99@gmail.com (Á.K.B.); nagybrigitta@oct.bme.hu (B.N.); gmarosi@mail.bme.hu (G.M.); hpataki@mail.bme.hu (H.P.); zsknagy@oct.bme.hu (Z.K.N.)

**Keywords:** residence time distribution, continuous powder blending, operation block model, feeder selection, funnel plot

## Abstract

The present paper reports a thorough continuous powder blending process design of acetylsalicylic acid (ASA) and microcrystalline cellulose (MCC) based on the Process Analytical Technology (PAT) guideline. A NIR-based method was applied using multivariate data analysis to achieve in-line process monitoring. The process dynamics were described with residence time distribution (RTD) models to achieve deep process understanding. The RTD was determined using the active pharmaceutical ingredient (API) as a tracer with multiple designs of experiment (DoE) studies to determine the effect of critical process parameters (CPPs) on the process dynamics. To achieve quality control through material diversion from feeding data, soft sensor-based process control tools were designed using the RTD model. The operation block model of the system was designed to select feasible experimental setups using the RTD model, and feeder characterizations as digital twins, therefore visualizing the output of theoretical setups. The concept significantly reduces the material and instrumental costs of process design and implementation.

## 1. Introduction

The pharmaceutical industry is traditionally batch-dominated, however, in recent years, a shift could be observed towards continuous technologies owing to their benefits such as more efficient quality control, better uniformity, and higher performance [1,2,3]. Case studies show that continuous technologies can significantly reduce the manufacturing costs (~40%) [1,4].

The introduction of continuous technologies was accelerated by the Process Analytical Technology (PAT, list of abbreviations is given at the end of the paper) framework proposed by the Food and Drug Administration (FDA), published in 2004 [5]. The framework recommends four PAT concepts to aid the development and quality assurance of pharmaceutical technologies. These are:Multivariate tools for design, data acquisition, and analysisProcess analyzersProcess control toolsContinuous improvement and knowledge management tools

Since then, many research groups have been working on implementing continuous technologies and a large body of literature was published in the scope of PAT concepts and process monitoring. However, these works mainly focus on in-line process analyzers, and multivariate tools needed for the analysis of the spectra, and they demonstrate only a few applications in control strategies [6,7].

In the past few years, numerous products manufactured in a continuous system have been approved by the FDA and European Medicines Agency (EMA). The FDA and International Conference on Harmonization (ICH) are working on guidelines for continuous manufacturing (CM) based on the approval of these products to encourage further development [8,9]. These recommend proper process monitoring with PAT tools, understanding the process dynamics, and developing advanced control strategies [8]. Modeling is an important task throughout the development of continuous technologies [9]. Measuring and modeling of the residence time distribution (RTD) represent a key part in the development, as RTD describes process dynamics [8] and can be used in the new aspects of continuous technologies. Publications address batch definition, and material traceability using the RTD of the system [10,11,12,13]. Research studies show the relation between the required sampling frequency for analytical measurements and the process dynamics [12]. An RTD-based control system was published demonstrating the advantages in waste management derived from the smoothing effect of the process [14], and RTD was used with feeder data as a soft sensor to predict active pharmaceutical ingredient (API) concentration and replace PAT tools [15].

During the production of solid dosage forms, most of the manufacturing lines include powder blending steps to mix the API with the excipients [16]. Continuous powder blending requires new equipment, because the particles have to blend as they flow through the appliance. Horizontal twin-screw blenders [17], and vertical blenders [18] are used to accomplish proper homogeneity. The ingredients are continuously fed into the blender with feeders, which can operate in volumetric or gravimetric modes.

Process analyzers are the core elements of the PAT framework. Process monitoring, achieved with in-line process analyzers, can be used for better process understanding, to model the process dynamics and to design control strategy [5].

Vibrational spectroscopic methods such as near-infrared (NIR) and Raman spectrometries [19,20,21] are widely used in solid formulation technologies [22,23], and they are the most frequently applied PAT process analyzers to monitor blend concentration. Reflectance probes can be installed into the production line as noninvasive, in-line process analyzers, and there is evidence to apply both spectroscopic methods for quantitative analysis in low dose formulations (<5% *w*/*w*) [24].

RTD modeling is the proposed way to describe process dynamics [8]. It has been used in the industry since 1958 for liquid applications [25], but it has a growing interest in solid formulations [26]. Two approaches have spread in most cases. The tank-in-series (TIS) model is created by describing the combination of continuously stirred tank reactors and plug flow reactors in series [27,28]. Axial dispersion model is a mechanistic model based on the Fokker–Planck equation and the Péclet number [29,30,31]. These two models are based on scientific approaches instead of statistical models, requiring less experiments for calibration to build robust models.

Flowsheet models were used to aid process design with in silico experiments [32,33]. The flowsheet model is an integrated model of the individual input and process models of the system. Using appropriate models enables determining the effect of process parameters with reduced material waste and to separate the sources of error, while the whole process is investigated.

Flowsheet modeling requires digital twin (DT) inputs of the feeders as mass flow-time datasets. Loss-in-weight (LIW) feeders can provide sufficient data. However, feeders may differ in data filtering and averaging, therefore catch scale experiments can be used to compare different feeders with the same data treatment. Gain-in-weight (GIW) measurements using a catch scale under the feeder were used to characterize feeding errors during operation and refill [34,35]. This method is applicable to build digital twins, because the mass flow fluctuations of the feeders are measured and can be used as a high-resolution input for simulations as it has the characteristics of the investigated feeder. Digital twins can be used to determine the feasibility of the feeder for the given process.

To utilize the advantages of CM over batch technologies control strategies have to be integrated into the process design. The products with unacceptable quality must be identified in a continuous process to eliminate from the further steps of the production line. Identification of the out of specification (OOS) materials can be achieved from the input data if the RTD model is known [14], or if the output is measured with PAT tools. RTD has a smoothing effect on the system that can be used to determine measurement frequency for in-line process analyzers and deviations in process parameters. Funnel plots can be used to visualize the maximum concentration upset caused by fluctuations from the concentration upset and the duration of the disturbance [33,36]. Tolerable and action limits with the two parameters appear as a funnel, showing the relation between required measurement frequency and precision. Funnel plots can be estimated from RTD models via simulation. 

In this study, a two-component continuous powder blending system was designed according to the PAT framework and CM guidelines. Acetylsalicylic acid (ASA) was mixed with microcrystalline cellulose (MCC) to produce a powder blend with 20% *w*/*w* API content for direct compaction with pharmaceutical grade content uniformity (CU). The examined process is a key part of a published fully integrated end-to-end concept, which was developed on this two-component system, with ASA as the model API [37]. The API content was monitored during the laboratory- and pilot-scale experiments by an in-line NIR probe positioned after the blender using partial least squares (PLS) modeling. The mass flow was limited by other pharmaceutical processes in the end-to-end concept, therefore, even the pilot scale experiments were under 1 kg/h total mass flow.

RTD was determined with impulse disturbances according to design of experiment (DoE) studies around two different mass flow scales to investigate the effect of the critical process parameters (CPPs) on the CU. RTD was described with two different models and techniques to achieve proper model fitness. The new technique for normalization during model building provided better description around the peak of the impulse response compared to the traditional normalization, which is beneficial for simulations. Operation block model was developed to select viable instrumentation. The models integrated the feeder deviations through catch scale digital twin characterization. Different feeder-based control strategies were tested to reduce waste via material diversion. This study intends to provide a thorough concept for process design applicable to many CM process in the pharmaceutical technology.

## 2. Materials and Methods

### 2.1. Materials

Acetylsalicylic acid (ASA) was used as a model API, and microcrystalline cellulose (MCC) was used as tableting excipient. Crystalline ASA (particle size: d_10_ = 133.16 µm; d_50_ = 532.86 µm; d_90_ = 1223.77 µm) is marketed as Rhodine^®^ 3040 (Novacyl, Écully, France). MCC (particle size: d_10_ = 113.88 µm, d_50_ = 247.93 µm, and d_90_ = 449.58 µm) is marketed as Vivapur^®^ 200 (JRS Pharma GmbH, Rosenberg, Germany).

### 2.2. Continuous Powder Blending Setup

Recently, we published an end-to-end continuous direct tableting line, involving a continuous powder blending step [37]. Powder blending is a key process preceding direct tableting, where mixing of the APIs and tableting excipients occur. In contrast to the batch technology, where premeasured amount of components is loaded into the blender, in a continuous powder blending setup, the API and the excipients are fed into the blender continuously and simultaneously, which impose an additional challenge on the feeding and blending process to ensure the adequate blend homogeneity and product content uniformity. Therefore, selection of feasible feeder setup is a crucial part of the process design. Axial and cross-sectional mixings occur when the materials are transported through the blender by screws. Axial mixing has a smoothing effect on feeding disturbances, which determines the action limits of the feeders and should be taken into consideration when the feeders are chosen.

The continuous powder blending of ASA and MCC was carried out by feeding the powders separately into a TS16 QuickExtruder^®^ (Quick 2000 Ltd., Tiszavasvári, Hungary) twin-screw multipurpose equipment [17,37,38] using a screw with 16 mm screw diameter (25 L/D ratio). The equipment is suitable for continuous melt extrusion, wet granulation, and homogenization. The screw (Figure 1b) contained feed screw and mixing elements. The configuration remained unchanged during the experiments.

A single-screw volumetric and two gravimetric feeders were tested to achieve proper feeding during the experiments. A conveyor belt carried the powder blend from the twin-screw blender to the collector plate. The blending can be coupled with continuous tableting with the conveyor belt [17,33], however, this was not utilized in this study. The blend was analyzed in-line with an NIR probe, mounted above the conveyor belt next to the blender in order to minimize the conveyor belt’s effect on the RTD. The belt transported the powder with constant speed of 1 cm/s. The experimental setup can be seen in Figure 1a.

### 2.3. Feeder Characterization

To reduce experiment time and material costs feeder characterizations were made. Three different feeders were tested for ASA feeding and two for MCC feeding before the powder blending experiments. A single-screw feeder (FPS Pharma, Fiorenzuola d’Arda, Italy) was tested with round geometry and low feeding volume that make it ideal for low mass flow rate feeding. Due to its limitations, it was only capable of the API feeding in the laboratory-scale experiments. The feeder can be controlled with serial connection, but it does not have an integrated scale or control system.

Two twin-screw feeders (Brabender loss-in-weight feeder, DDW-MD0-MT-1.5 HYD with Congrav OP 1T operator interface and a MechaCAD gravimetric feeder) were tested for the ASA and MCC feeding. The screws of the feeders have big free volume and oval geometry (MechaCAD: 10.2 mm × 12.8 mm; Brabender: 9.0 mm × 11.5 mm) as shown in Appendix A. The twin-screw feeders can be used for volumetric and gravimetric feeding.

The characterization was carried out using a gain-in-weight (GIW) scale setup. An analytical scale (Sartorius L420P) was placed under the feeder as catch scale. With this setup, each feeder could be tested with the same, high precision scale, because the feeder is not on the scale as in a loss-in-weight (LIW) setup. The data from the scale was collected by an in-house developed software to record the weight change in time, during the feeding experiments. Mass flow was evaluated from the weight change, resulting the feeder inputs as a digital twin of the feeder.

### 2.4. NIR Spectrometry and Data Analysis

The blend was analyzed with a Bruker MPA FT-NIR spectrometer (Bruker Optik GmbH, Ettlingen, Germany) equipped with a Solvias fiber optic probe (Solvias AG, Kaiseraugust, Switzerland) in reflection mode. The probe was placed above the conveyor belt, at the die of the twin-screw blender to obtain real-time spectra of the powder blend with minimal time delay. NIR spectra were collected with 8 cm^−1^ resolution at the range of 4000–12,500 cm^−1^ by accumulating 16 scans in the case of calibration measurements. In order to reduce measurement time and achieve the most information of the process during in-line monitoring, the number of scans was reduced to 4 resulting fast measurement (<1 s), and proper measurement frequency (3.5 s/spectrum). Less scans would result in poor spectra without improving the measurement frequency as the time between two spectral acquisitions is constant.

Spectrum accumulation was completed using OPUS^®^ 7.5 software (Bruker). Evaluation of the spectra was carried out by MATLAB 9.7 (MathWorks, Natick, MA, USA) and PLS Toolbox 8.7.1. (Eigenvector Research, Manson, WA, USA). A graphical user interface (GUI) was developed in-house with MATLAB scripts, to achieve the real-time evaluation of the acquired spectra.

The ASA content of the blend was determined from the spectra via PLS regression. The aim of the model building was to achieve a robust model that can be used in the real-time measurements of the moving powder blend with high sampling frequency. Furthermore, a sampling method was developed for tablets to replace destructive off-line API content measurement with an at-line nondestructive NIR method using the same probe and evaluation model as the in-line procedure.

For the model building, a calibration set (11 samples) was used in the 0–100% *w*/*w* ASA concentration range by blending 500 g of ASA and MCC per 10% in bottles by manual shaking for 5 min. The blends were placed on the conveyor belt modeling the in-line measurement setup and the powder in motion was analyzed with the NIR spectrometer. This method prevented the bias originating from the big crystals as the moving powder blend provided bigger sample volume, thereby a more representative sampling.

### 2.5. Determination of Residence Time Distribution

RTD of the continuous powder blending was determined by impulse disturbances. MCC was fed to the twin-screw blender with the Brabender twin-screw feeder. The feeder was used in gravimetric mode, set to the examined total mass flow. The powder transport was fast enough in the powder blender with the chosen rotational speeds to avoid any accumulation at the hopper as that would have affected the residence time distribution.

ASA was introduced to the system by manually adding pure ASA directly into the hopper of the blender. Different impulse sizes were tested in the range of 0.5–2.5 g ASA (1.5 g). To determine the size of the disturbance on the system, the mass hold-up was calculated from the *MRT* of the system and the mass flow with Equation (1) and the spike was compared to the hold-up with Equation (2).
(1)$$m_{hold\text{-}up}=MRT\times \dot{m},$$
(2)$$d=\frac{m_{impulse}}{m_{hold\text{-}up}}\times 100\%$$
where *m_hold-up_* is the hold-up mass in the blender, *MRT* is the mean residence time, $\dot{m}$ is the mass flow, *d* is the disturbance rate, and *m_impulse_* is the mass size of the impulse disturbance.

The impulse response was evaluated with NIR spectrometry after the powder blender. The time period between the spectral acquisitions was intended to be reduced in order to obtain the most amount of information about the RTD of the system. For this purpose, we used only four scans, after that the acquisition time was determined mainly by the waiting time between two measurements, so further reducing the number of scans does not result in significantly higher sampling frequency. NIR spectra were evaluated in real-time and repeated impulse disturbances were made when the ASA concentration settled at zero.

The effect of the CPPs on the blending (mass flow and screw rotation speed) was studied using DoE studies. The mass flow in general is determined by the equipment’s capacity and the capacity need. The minimum screw rotational speed is determined by the mass flow, as the screw rotation has to be capable of transporting the material without accumulation. These two factors were changed at two levels according to a two-factor 22 experimental design. In order to study the scale-up of the system, two different mass flow ranges were examined with two sets of DoEs. Three repetitions were made at each experimental setting. The main parameters of the distribution were evaluated by numerical integration of the responses according to Equations (3)–(5), and the effects of the factors were determined with ANOVA tests.
(3)$$MRT=\int_{0}^{\infty }t\times E\left(t\right)dt,$$
(4)$$Var=\int_{0}^{\infty }{\left(t-MRT\right)}^{2}\times E\left(t\right)dt,$$
(5)$$\sigma =\sqrt{Var},$$
where *E(t)* is the probability distribution function (PDF) of the process in *t* as time, *Var* is the variance of the PDF, and *σ* is the standard deviation of the PDF.

### 2.6. Residence Time Distribution Modeling

Impulse disturbances were used on the continuous powder blending setup to measure the RTD of the blender.

The modeling was done with the least squares method after preprocessing of the measured responses. The system PDF was defined by Equation (6) for impulse disturbances. The preprocessing consisted of noise reduction before and after the impulse response and numerical normalization of the response by area under curve (AUC), according to Equation (7). The sum-of-square error is calculated according to Equation (8).
(6)$$E\left(t\right)=\frac{c\left(t\right)}{\int_{0}^{\infty }c\left(t\right)dt},$$
(7)$$c_{n,AUC}=\frac{c\left(i\right)}{\sum_{i=1}^{n-1}\left[\frac{c\left(i\right)+c\left(i+1\right)}{2}\right]\times \left(t_{i+1}-t_{i}\right)}$$
(8)$$SSE=\sum_{i=1}^{n}{\left[{E\left(t\right)|}_{t_{i}}-c_{n}\left(i\right)\right]}^{2},$$
where *c*(*i*) is the measured concentration in *t_i_* time, $c_{n,\mathrm{method}}\left(i\right)$ is the normalized concentration in *t_i_* time with maximum or *AUC* normalization method, *n* is the number of concentration measurement points, *SSE* is the sum-of-square error, and ${E(t)|}_{t_{i}}$ is the calculated probability distribution function in *t_i_* time.

The impulse responses had a significant tail that affected the aforementioned method. The tail has high impact on the model building, therefore the models could not properly describe the peak of the distribution. We have introduced a new method for model building, where we normalized the concentration response and the PDF according to the peak maximum with Equation (9). After calculating the model parameters, the model was normalized by its AUC to retrieve the actual PDF of the system.
(9)$$c_{n,max}=\frac{c\left(i\right)}{max\left(c\right)}.$$

Tank-in-series (TIS) and axial dispersion (AD) models were used for RTD modeling.

TIS models were developed by Equations (10) and (11) using the combination of plug flow models and multiple same sized continuously stirred tank reactors. The model bases on three parameters. *MRT* is the mean residence time of the model, initially calculated from the impulse response, but adjusted during the model fitting, while dead-time and the number of used tanks can be adjusted.
(10)$$E\left(t\right)=\frac{N\times e^{-\frac{t-t_{D}}{\tau_{Tank}}}\times {\left(\frac{t-t_{D}}{\tau_{Tank}}\right)}^{N-1}}{\tau_{TIS}\times \left(N-1\right)!},$$
(11)$$\tau_{\mathrm{Tank}}=\frac{\tau_{\mathrm{TIS}}-t_{D}}{N},$$
where *t_D_* is the dead-time, *N* is the number of tanks used in the model, *τ_TIS_* is the *MRT* of the model, and *τ_Tank_* is the *MRT* of one tank.

Axial dispersion models were developed by Equation (12), with open–open boundaries. The model has two parameters, the *MRT* and the Péclet number.
(12)$$E\left(t\right)=\frac{1}{\tau }\sqrt{\frac{Pe}{4\pi \left(\frac{t}{\tau }\right)}}\times \mathrm{Exp}\left[-\frac{\mathrm{Pe}\times {\left(1-\frac{t}{\tau }\right)}^{2}}{4\times \frac{t}{\tau }}\right],$$
(13)$$Pe=\frac{\dot{m}\times l}{E},$$
where *τ* is the *MRT* of the model and *Pe* is the dimensionless Péclet number, which can be derived from the $\dot{m}$ mass flow, *l* length, and the *E* dispersion coefficient with Equation (13).

### 2.7. Operation Block Model

The total powder blending setup was modeled with operation block modeling assembling the feeder characterizations and the RTD model. This model can be used to build flowsheet models of systems that contain this operational unit. Figure 2 shows the simplified scheme of our model. The measured mass flow data with fixed measurement window is the digital twin of the feeder. Feeder inputs are combined resulting in a concentration input for the simulation. The process dynamics in the powder blender is modeled with the RTD. The output of the simulation is evaluated from the convolution of the concentration input and the PDF of the RTD. The operation block model can be used for a feasibility filtering with systematic mapping with the feeder digital twins and process models.

## 3. Results and Discussion

### 3.1. ASA Content

The PLS regression model allows us to quantify the ASA content in the blend at the full concentration range (0–100%). First, the measured spectra were truncated to 7850–4300 cm^−1^, where the components absorbed the NIR light significantly. Spectra from triplicated measurements were pretreated by standard normal variate (SNV) and mean centering before PLS regression model building. Root mean square error (RMSE) was used for indicating the goodness of fit. Using 3 latent variables (LVs), the examined values were 1.27 for the calibration (RMSEC) and 1.34 for the cross-validation by random subsets (RMSECV). The R^2^ was over 0.997 (Figure 3).

### 3.2. Residence Time Distribution

During the optimization of the impulse sizes 0.5 and 1.5 g ASA impulses were tested. In the pilot-scale experiments, the lower impulse sizes were near to the limitations of the NIR measurement. In the laboratory-scale experiments, the peak was similar with each impulse size, but the 0.5 g impulse failed to capture the tail of the distribution (Appendix A). Therefore, 1.5 g impulses were chosen for further experiments to achieve proper signal to noise ratio (S/N).

In the first DoE, a laboratory-scale mass flow setup was studied, with a central point at 0.27 kg/h mass flow and 50 rpm screw rotational speed. The mass flow was changed by 0.03 kg/h in each direction and screw rotational speed was tested on 40 rpm as the lower and 60 rpm as the higher level. Three replicates were done in each setting. The settings of the experiments are summarized in Table 1. According to the ANOVA test results and the effect plots shown in Figure 4, only screw rotational speed had significant effect on the *MRT*, the *σ* of the RTD was independent from the investigated CPPs in the experimental range. This is in correspondence with the results reported by Vanarase and Muzzio [39].

The second DoE was carried out using a pilot-scale setup at around 0.9 kg/h mass flow and 90 rpm rotational speed. The mass flow was adjusted between 0.8 and 1.0 kg/h, screw rotational speed was changed from 80 to 100 rpm. The results shown in Table 1 reveal the same relation between the parameters as the laboratory-scale experiments but the effect of screw rotational speed on *MRT* is lower.

The effects of the two main factors were tested by changing one factor at a time, using a wider range compared to the experimental designs. The experiments are summarized in Appendix A, and the effect plots are shown in Figure 5. The results match the relations shown by the two DoE studies but in this wider range, the mass flow has a narrowing effect on the PDF (Appendix A).

According to the ANOVA tests, the screw rotational speed has a significant effect on the *MRT*, however, it has no dependence on the mass flow. The ANOVA tests suggest that the mass flow affects *σ*, and it slightly changes in the wider range experiments, but in normal operating conditions, it does not depend on the disturbances of the rotational speed.

The mass hold-up was the smallest in the 0.24 kg/h mass flow 60 rpm screw rotational speed setup, where the estimated hold-up was 4.84 g, which means that d, the rate of the impulse compared to the hold-up, was 30.99% at the maximum, which is comparably high to the mass hold-up, but as the study shows, the mass flow has little impact on the system dynamics and the simultaneous 0.5 g tracer experiments presented similar curves.

The two different models and preprocessing are applied to each experimental setting. The model parameters for each experimental setup are summarized in Table 2 and Table 3 with two different normalization methods for model fitting, accounting the maximum of the curve and the area under curve (AUC). The parameters differ slightly with the two model fitting techniques.

The parameters of the fitted TIS models suggest that the system is dead time dominant as dead time takes up from 66.8% to 83.6% of the *MRT*. Nevertheless, it is worth mentioning that the number of tanks used in the models are rather low, ranging between 1 and 15. In the TIS model, increasing the two parameters, *N* and *t_D_*, simultaneously narrows the distribution.

The AD models have high *Pe* numbers in a wide range (107–328), which is also caused by the high dead time in the system. The models with lower *Pe* numbers correlate with the dead time ratio in the TIS model with the only exception in the 0.80 kg/h, 100 rpm setup, but in that case, part of the dead time is described by the high (15) number of tanks used in the TIS model.

The differences between the two used models are presented in Figure 6a. The models describe the distribution properly, but slight differences can occur between the two models. The two normalization methods result in similar differences between the best models as using different modeling equation as it is shown in Figure 6b. According to these experiences, data pretreatment is just as important during model building as the selection of the model.

To quantify model goodness over the usual indexes (RMSE and sum of square error), we have introduced an error plot that presents the model error in time as it is shown in Figure 6c,d. The portion of absolute errors are plotted against time; therefore, the source of error is highlighted. Ideally it is constant, because the noise is present and constant at every measurement point. At one selected setting, using normalization based on peak maximum, Figure 6c depicts that the error is zero at the peak maximum and describes the peak properly, but the error is higher at the tail of the distribution. Figure 6d depicts the differences between the used normalization methods. Using the normalization based on AUC, the error at the tail is reduced but most of the error originates from the descending part of the peak.

Normalization to peak maximum is independent of the measurement noise at the tail. However, the AUC method reduces noise as the noise offsets over time. The calculated errors are shown in Figure 7a,b using the different methods. The figure presents how changing the two parameters of the AD model affects the sum of squares that the models are classified by. Each contour on the figure represents 25% increased difference in the sum of squares compared to the best model. The shape of the contours and the minima calculated by the two methods differ significantly.

### 3.3. Feeder Characterization and Feeder-Based Process Control

The integrated operation block model requires feeders as inputs. For this purpose, digital twins of the feeders have to be created via feeder characterization from the RAW data. The RAW data is exported from the scale with the highest possible resolution; therefore, many points were registered where no weight change is measured. The digital twins simulate the characteristics of the feeding process and can be used for in silico experiments. Furthermore, these analytical solutions can be used to build soft sensors from the feeders, providing a fast and reliable monitoring and control tool for the process. The characterization was carried out with catch scale experiments.

To build digital twins of the feeder the time resolution must be set as it is a requirement for further mathematical evaluations. The sampling time of the raw data would result in unreliable mass flow data, as the time differences are not uniform and mass change does not occur in most measurement points. Choosing proper time resolution is crucial as it is presented in Figure 8 on a catch scale experiment with the single-screw feeder with a set point of 1 g/min (0.06 kg/h) ASA feeding to achieve pharmaceutical grade CU (5% action limits) for laboratory-scale (0.30 kg/h) powder blending. If the resolution is too high, the precision of the calculated mass flow is reduced, and the differences seem higher than reality. However, if the mass flow is calculated from large time intervals, small but significant disturbances can be overlooked.

To address this problem, the time resolution has to be adjusted to the process dynamics. If the resolution is calculated from the RTD, then every disturbance that affects the system on an unacceptable level can be noticed, but small or fast disturbances does not exceed action limits. Multiple approaches are available to utilize the smoothing effect of the residence time distribution on the feeding data. The measurement frequency of every analytical sensor or, in this case, soft sensor should provide 3–5 measurement points over 2*σ* period of the RTD.

Funnel plots are commonly used to visualize feeder upsets. Two different funnel plots are shown in Figure 9, generated from the same experimental data with the single-screw feeder. On the funnel plots, the error from the set point, the mass flow or concentration upset is plotted against the duration of the upset. Every point of the plot presents the maximum deviation in the product caused by the measured disturbance. Action limits can be presented as contour plots approaching the corresponding mass flow upset but widening at low durations forming a funnel. The 5% action limit for the powder blending process calculated from the RTD is plotted with solid red lines. If a measured disturbance is placed outside of the action limit, then the disturbance causes waste. However, placement of a disturbance on the funnel plot may not be obvious if the data varies significantly, seemingly lacking trends. The funnel plots presented in Figure 9 show two different approaches.

In Figure 9a, the average mass flow upset is calculated from every point and for various time periods. The disturbances are then marked with the color representing the time point it is calculated from. That way the plot can be used real time and the deviances can be located in time, but trends for the whole run are only shown in point density.

Another approach is presented in Figure 9b. After the deviances are evaluated, the plot is divided into smaller areas and the relative frequency is calculated. The whole experimental range is presented on the plot with the colors presenting the density of the disturbances with given amplitude and duration. With this approach, trends during the experiments can be identified. In the displayed dataset, the points are shifted from the set point, showing a shift in the feeding for a longer time period.

RTD can be used to calculate the output function from high-resolution or even raw input feeder data. Bhaskar and Singh [14] proved that using RTD-based control systems are most beneficial with data containing random disturbances, in which the action limits are violated frequently in each direction. The product may be acceptable after the blender, but the violations initiate the material diversion protocol in conventional control strategies. High-resolution (<2 s) mass flow data provide such dataset as input for the blender.

If the concentration is calculated by convolving the inputs from the feeders and PDF of the system, the concentration upset in the output can be evaluated. That way, a smoothed concentration profile can be achieved, and the quantity of out-of-specification product can be calculated. Figure 10 demonstrates the difference caused by the process dynamics. The same mass flow data was used with different RTD models. The model for the 0.3 kg/h mass flow and 60 rpm screw rotational speed is less sensitive, as it can smooth out bigger mass flow fluctuations, and the calculated concentration barely exceeds the 5% limit in accordance with the disturbances seen in the 10 s resolution input data (Figure 8). However, at one point, the input data exceeds the limit at 1.5 min (90 s), but this is a fast disturbance that the back mixing can balance. The output concentration calculated from the 1 kg/h 100 rpm model exceeds the 5% limit more often, and some of these could not be shown if the mass flow is presented in 10 s resolution.

The two twin-screw feeders were tested for the two materials on the laboratory- and pilot-scale setups. The single-screw feeder has low capacity; therefore, it was only tested for the laboratory-scale ASA feeding.

The single-screw feeder tested for ASA feeding had a maximum mass flow of 1.44 g/min (0.0864 kg/h), and it was set to 1.00 g/min (0.0600 kg/h) with volumetric feeding for the laboratory-scale experiments. Based on the catch scale experiments, the mass flow had a high variance at a frequency of 1 s, but it was stable. The two twin-screw feeders have higher capacities, and they are limited in the low mass flow range experiments. As the feed rate of ASA was reduced to 1 g/min (0.06 kg/h), the mass flow became periodic and low frequency peaks appeared because of the screws’ uneven geometry as it can be seen in Figure 11.

The three 1 g/min feeding data were compared after convolution with the RTD model of the laboratory-scale setup (0.3 kg/h mass flow and 60 rpm screw rotational speed). The two twin-screw feeders cannot feed the ASA in such low quantities, with the MechaCAD producing uncontrolled feeding and the Brabender showing high disturbances. In such low quantities, only the single-screw feeder produced proper content uniformity (see Appendix A).

### 3.4. Operation Block Model

The control strategies only considered the effect of the feeders separately on the powder blending system. The block model integrates the inputs of the system and the process model to evaluate the output content uniformity of the system. The digital twin models of the feeders as inputs and the RTD model of the powder blender as process model produce the integrated operation model of the system. The output of the operation block model is the simulated product concentration in time. The model can be used to facilitate process design and investments, as unit models are constructed separately, therefore reducing the need for device and work time and setups inherently infeasible can be eliminated from experiments. 

Feeder digital twins and the RTD models were changed to find capable laboratory- and pilot-scale configurations to produce blends with pharmaceutical grade content uniformity. Multiple feasible configurations could be chosen, but many configurations produce insufficient content uniformity (Figure 12). Laboratory-scale experiments had less feasible options as only the single-screw feeder is capable of feeding ASA at low mass flow rates.

## 4. Conclusions

This work presents a thorough process design of a continuous powder blending process of a 20% *w*/*w* ASA and MCC blend with content uniformity of pharmaceutical quality. The aim of the present research was to examine the importance of process dynamics and modeling during design, monitoring, and control of the system. The results show that process dynamics are an essential part of the design.

The RTD of the given system was examined by changing the total mass flow and the screw rotational speed on a wide design space to describe the effects of these parameters. Tracer experiments on two DOEs and changing one factor at a time on expanded Design Space proved that even small changes in screw rotational speed determine *MRT*, but on a wider scale, the mass flow affects the width of the PDF. The distribution was modeled using the two most frequently used RTD modeling techniques, axial dispersion and tank-in-series models, and LS model fitting. We have implemented a new normalization technique for model building, by dividing the impulse response and the PDF by its peak maximum before the calculation of the *SSE*. Using this technique provided models with a better fit near to the peak, but worse at the tail of the system. These models were beneficial in the later simulations, as the simulations depend on the peak, rather than the tail.

Process monitoring PAT tools are essential in continuous manufacturing. NIR spectrometry proved to be a reliable monitoring tool for ASA content during operation by providing fast sampling time even compared to the process dynamics. The spectra were evaluated in real-time via PLS regression. The results indicate that the API can be used to monitor impulse responses and determine RTD. In tracer experiments the selection of the tracer is crucial, as the material properties affect the flow dynamics of the system. By applying API impulses, no extraneous tracer molecules were introduced into the system, which reduces the chance of changed flow dynamics due to the tracer. On the other hand, the modeling is limited by the goodness of the NIR model, and the homogeneity of the powder if the impulse is measured in the final concentration.

The RTD models were investigated and utilized for many purposes. Measurement frequency of process analyzers and feeder scales can be optimized based on the variance of the distribution. The models were used to create process control strategies based on feeding data, with funnel plots and advanced material traceability considering the smoothing effect of the axial mixing in the equipment and the feeder data as soft sensor to measure content uniformity. An operation block model was developed using digital twins of the feeders and RTD of the system with different configurations to aid process design through modeling.

This concept reduces the cost during research and operation. The optimization via RTD models reduces the material cost, while the optimization of the feeder setup can reduce investments during the research phase. Furthermore, the presented control strategy reduces the cost of operation, by waste management. The cost reduction is most significant when expensive API is used.

## Figures and Tables

**Figure 1 pharmaceutics-12-01119-f001:**
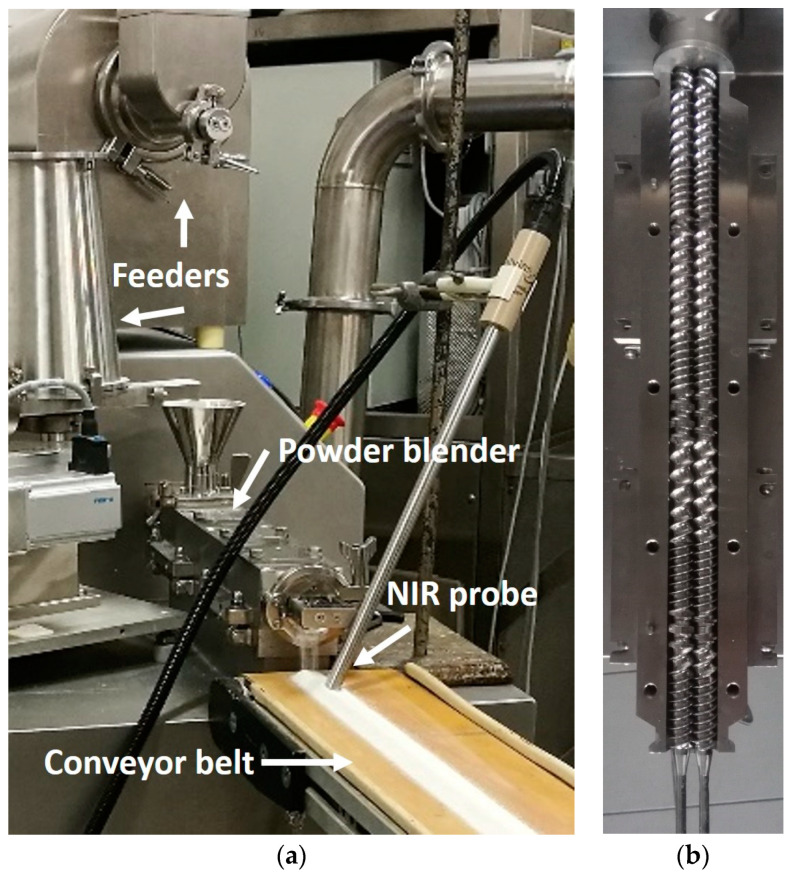
The used devices for experiments: (**a**) experimental setup with two feeders, the powder blender, and the NIR-spectrometer and (**b**) screw configuration description.

**Figure 2 pharmaceutics-12-01119-f002:**
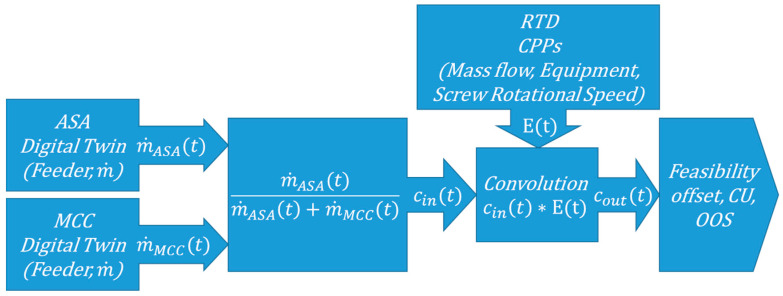
Operation block model of the system ($\dot{m}(t)$: mass flow in *t* time point and *c*(*t*): concentration in *t* time point).

**Figure 3 pharmaceutics-12-01119-f003:**
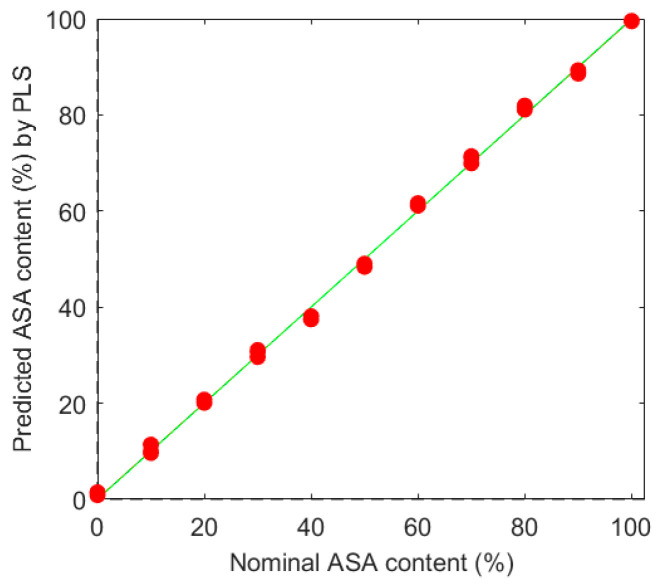
Calibration points by built partial least squares (PLS) regression method.

**Figure 4 pharmaceutics-12-01119-f004:**
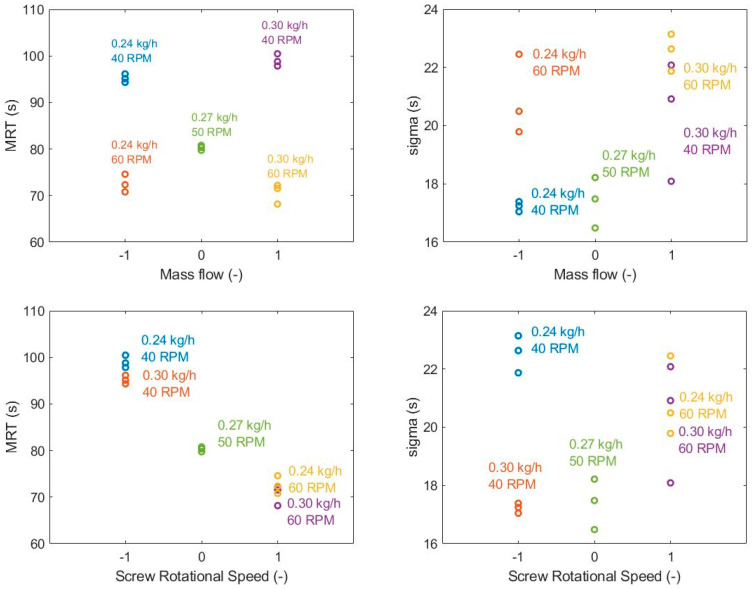
Effect plots of the two changed process parameters and the residence time distribution (RTD) parameters in laboratory-scale experiments with a central point of 0.27 kg/h mass flow rate and 50 rpm screw rotational speed.

**Figure 5 pharmaceutics-12-01119-f005:**
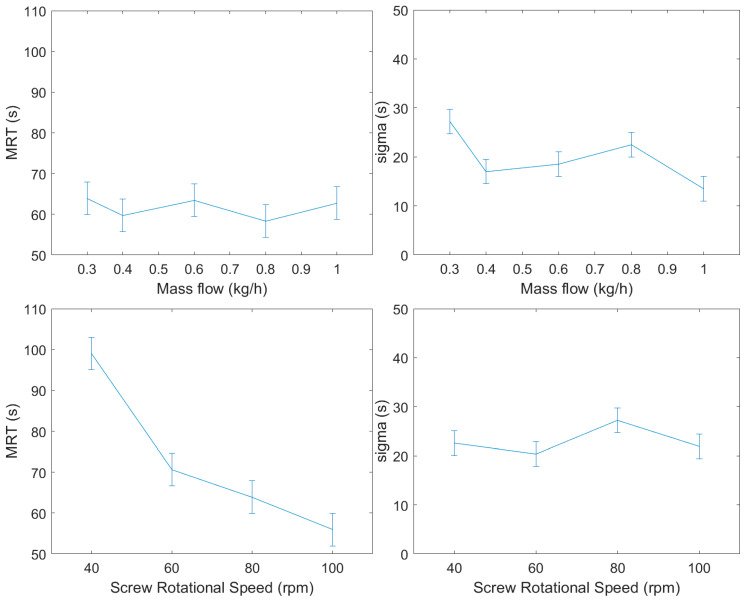
Effect of mass flow on the mean residence time and *σ* using one factor at a time in experiments.

**Figure 6 pharmaceutics-12-01119-f006:**
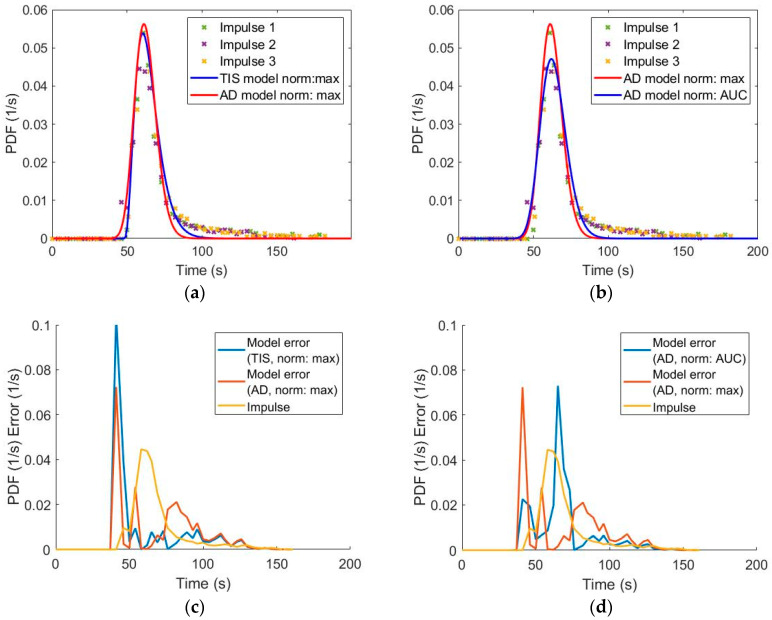
Model fit differences on the 0.30 kg/h, 60 rpm experiment: (**a**) fit of tank-in-series (TIS) vs. axial dispersion (AD) models normalized to peak maximum, (**b**) fit of AD models built with normalization to peak maximum vs. area under curve (AUC), (**c**) source of error of TIS and AD models normalized to peak maximum, and (**d**) source of error of AD models normalized to peak maximum vs. AUC.

**Figure 7 pharmaceutics-12-01119-f007:**
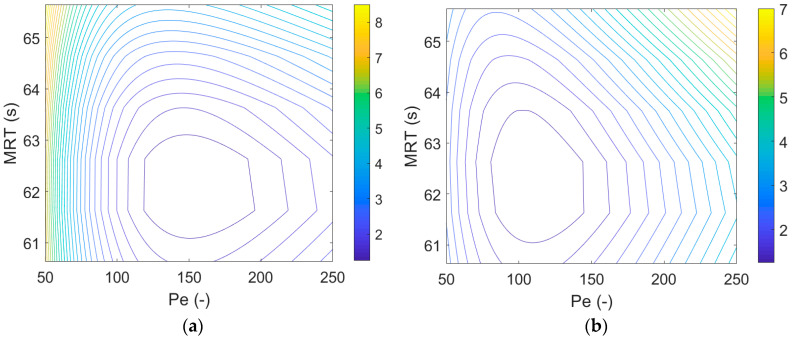
Effect of the model parameters on the sum of square error with different normalization methods: (**a**) contour plot of AD model fit calculated from normalization to peak maximum and (**b**) contour plot of AD model fit calculated from normalization to AUC.

**Figure 8 pharmaceutics-12-01119-f008:**
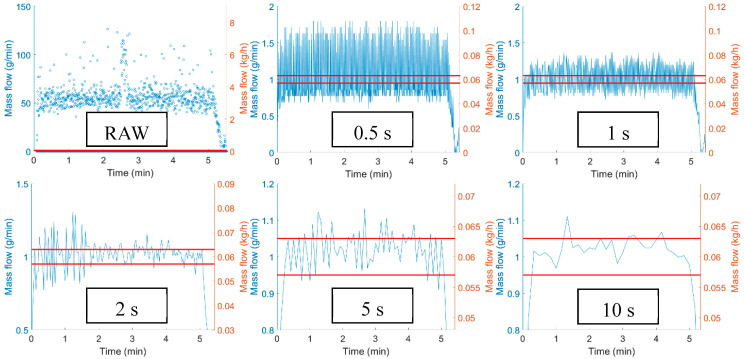
Feeder characteristics of a single-screw feeder with different sampling frequency (raw to 10 s), where red lines represent the 5% specification limits for 1 g/min set point.

**Figure 9 pharmaceutics-12-01119-f009:**
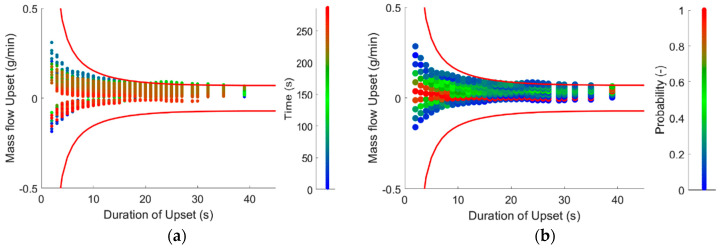
Funnel plots using the single-screw feeder to achieve 1 g/min feeding with 5% action limits: (**a**) disturbances placed with color representing the time when the disturbance started and (**b**) distribution of disturbances on the funnel plot.

**Figure 10 pharmaceutics-12-01119-f010:**
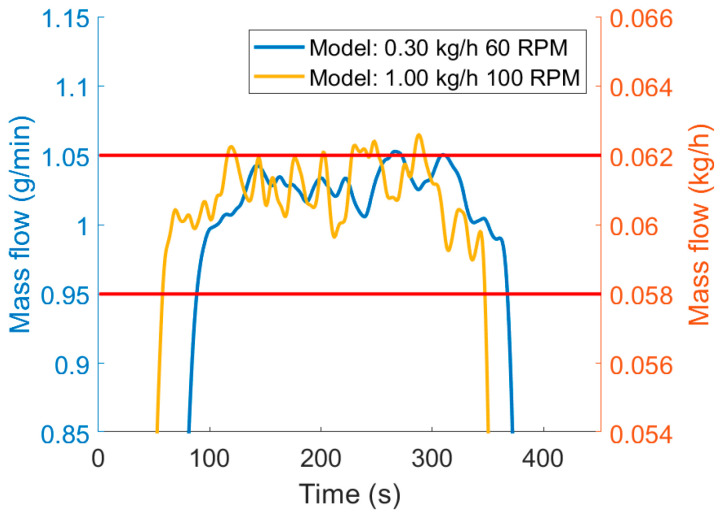
Broad RTD (blue) reduce waste compared to tighter distribution (yellow).

**Figure 11 pharmaceutics-12-01119-f011:**
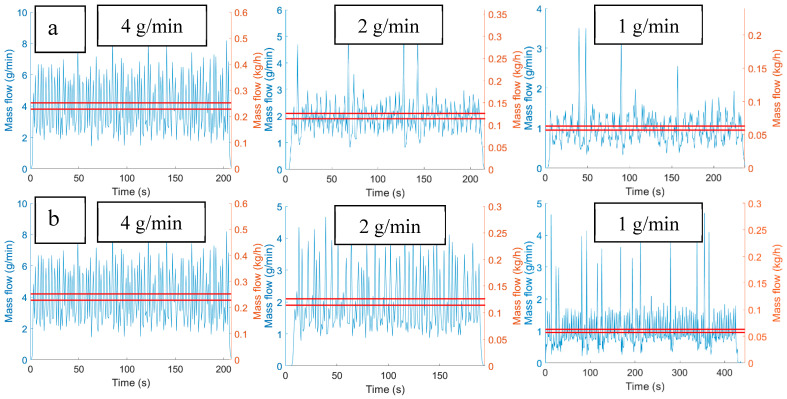
Effect of reduced mass flow on twin-screw feeder profiles: (**a**) MechaCAD feeder with 4, 2, and 1 g/min feeding and (**b**) Brabender feeder with 4, 2, and 1 g/min feeding.

**Figure 12 pharmaceutics-12-01119-f012:**
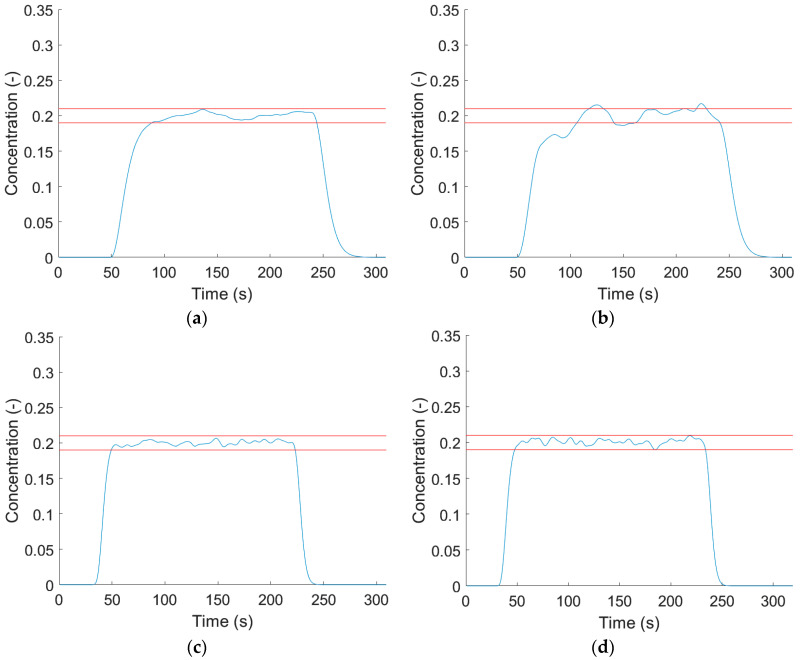
Content uniformity result of operation block model at pilot scale: (**a**) ASA-single-screw and MCC-MechaCAD, laboratory-scale; (**b**) ASA-Brabender and MCC-Brabender, laboratory-scale; (**c**) ASA-MechaCAD and MCC-MechaCAD, pilot-scale; and (**d**) ASA-MechaCAD and MCC-Brabender, pilot scale.

**Table 1 pharmaceutics-12-01119-t001:** Results of the two 22 experimental designs on the laboratory-scale and pilot-scale setups and the corresponding residence time distribution (RTD) parameters. “-” means dimensionless, normalized, between −1 and 1.

	Mass Flow		Screw Rotational Speed		*MRT*	*σ*	$m_{hold\text{-}up}$
	(-)	(kg/h)	(-)	(rpm)	(s)	(s)	(g)
Laboratory scale	1	0.30	1	60	70.63	20.36	5.89
	1	0.30	−1	40	99.03	22.55	8.25
	−1	0.24	1	60	72.57	20.91	4.84
	−1	0.24	−1	40	95.19	17.23	6.35
	0	0.27	0	50	80.29	17.39	6.02
Pilot scale	1	0.80	1	100	48.85	22.42	10.86
	1	0.80	−1	80	58.34	22.47	12.96
	−1	1.00	1	100	46.84	18.87	13.01
	−1	1.00	−1	80	62.74	13.48	17.43
	0	0.90	0	90	60.69	18.59	15.17

**Table 2 pharmaceutics-12-01119-t002:** Tank-in-series (TIS) model parameters based on Least Squares (LS) model fitting with normalization to peak maximum and area under curve.

Mass Flow (kg/h)	Screw Rotational Speed (rpm)	Maximum			AUC		
		*MRT* (s)	*N* (#)	$t_{D}$ (s)	*MRT* (s)	*N* (#)	$t_{D}$ (s)
0.24	40	92.19	3	73	92.19	4	68
0.24	60	71.57	1	51	67.57	4	44
0.27	50	76.29	2	59	77.29	3	58
0.30	40	92.03	6	69	94.03	4	70
0.30	60	64.63	4	48	65.63	3	48
0.30	80	53.89	5	36	54.89	4	36
0.30	100	47.93	4	34	49.93	3	34
0.40	80	55.72	3	43	56.72	3	42
0.60	80	57.45	3	48	59.45	1	50
0.80	80	50.34	9	37	51.34	5	38
0.80	100	41.85	15	29	41.85	9	30
0.90	90	54.69	6	43	55.69	5	42
1.00	80	62.74	1	50	61.74	2	48
1.00	100	39.84	9	28	40.84	4	30

**Table 3 pharmaceutics-12-01119-t003:** Axial dispersion (AD) model parameters based on LS model fitting with normalization to peak maximum and area under curve. Péclet (*Pe*) number is dimensionless.

Mass Flow (kg/h)	Screw Rotational Speed (rpm)	Maximum		AUC	
		*MRT* (s)	*Pe* (-)	*MRT* (s)	*Pe* (-)
0.24	40	88.19	172	89.19	129
0.24	60	63.57	113	64.57	79
0.27	50	73.29	164	73.29	133
0.30	40	90.03	222	90.03	139
0.30	60	61.63	151	62.63	109
0.30	80	50.89	114	51.89	74
0.30	100	45.93	107	45.93	73
0.40	80	53.72	149	53.72	115
0.60	80	55.45	328	56.45	206
0.80	80	59.34	276	49.34	180
0.80	100	40.85	326	40.85	219
0.90	90	53.69	302	53.69	195
1.00	80	56.74	231	57.74	104
1.00	100	38.84	217	39.84	125

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
