# Peer review of "Process Design of Continuous Powder Blending Using Residence Time Distribution and Feeding Models"

_pharmaceutics, 2020, doi:10.3390/pharmaceutics12111119_

Round 1

Reviewer 1 Report

The submitted paper is focused on the overall presentation (design, control, process dynamics) of a typical pharmaceutic operation: fine materials blending. The particularity of the performed research activity is in the realization of a continuous (and efficacious) blending operation of powders (acetylsalicylic acid and microcrystalline cellulose were used as model ingredients; ASA as tracer).

In my opinion the topic is of great interest towards the modernization of the pharmaceutical industry and my global evaluation of the submitted paper is positive.

However, I judge the manuscript difficult to read for the sum of specific aspects related to the process dynamics approach and results representation. My suggestions are addressed to present (also via pictorial images) the pathway of the performed studies reporting at the end of the Introduction section (aims) and in Conclusions, more explicative sentences about problems / criticisms to overcome; achieved experimental evidences (bad and good) for less expert Readers. Moreover:

- where is possible to limit the use of acronyms (a list could help the readers);

- info about size and size distribution of used powders should be reported;

- consideration about possible segregation phenomena should be added.

Due these kinds of comments s (text reformulation, info adding) I think that "minor revision" is appropriate as final score.

Please delete sentences at page 3 lines 115- 126 (template recommendations).

Author Response

However, I judge the manuscript difficult to read for the sum of specific aspects related to the process dynamics approach and results representation. My suggestions are addressed to present (also via pictorial images) the pathway of the performed studies reporting at the end of the Introduction section (aims) and in Conclusions, more explicative sentences about problems / criticisms to overcome; achieved experimental evidences (bad and good) for less expert Readers.

Answer: The aims have been filled out in the introduction to provide a better understanding and in the conclusions. (page 3 lines 104-109; lines 112-115; page 17 lines 475-479; lines 481-482; line 489; lines 494-498)

Moreover:
- where is possible to limit the use of acronyms (a list could help the readers);

Answer: A list of abbreviations has been added (lines 522-568).

- info about size and size distribution of used powders should be reported;

Answer: The particle size distribution has been described in the materials section by the d10, d50 and d90 parameters of the ASA and MCC according to Malvern Mastersizer 2000 measurements. (page 3 lines 123-125)

- consideration about possible segregation phenomena should be added.

Answer: In the close future, we plan to study the effect of particle size in the Residence Time Distribution, but it was not in the scope of this study. In this study, we have experimented with a commercially available API lot.

Please delete sentences at page 3 lines 115- 126 (template recommendations).

Answer: They have been deleted.

Reviewer 2 Report

This manuscript addresses an intensified study on a continuous powder blending process design of acetylsalicylic acid (as the model drug) and microcrystalline cellulose (as the model excipient) based on Process Analytical Technology. The in-line process monitorization was leveraged through the use of multivariate data analysis.

The manuscript is well written and the design of the study has been well prepared; indeed, it investigates in a detailed and systematic manner the processing conditions involved and impacting the blending process.

I would recommend this to be published at Pharmaceutics, after the authors address some minor comments listed below.

Abstract

The final sentence in this section states that “The concept significantly reduces the material and instrumental costs of process design and implementation”. Can the authors provide a quantitative figure on this claim based on their study presented in this manuscript?

Introduction

Good overview of the various topics linked to continuous manufacturing, PAT, modelling but the rationale for selecting ASA as the API model system is not provided.

Materials and Methods

- There are three paragraphs in the beginning of the section which are a copy/paste from the Journal’s guidelines and should not be there.

- Characteristics for the raw ASA and MCC used are missing (e.g. solid-state/polymorphic form for ASA, particle size and size distribution for ASA and MCC, etc.)?

Results

The pilot-scale experiments (Table 1) do not use a significantly higher rpm, should the authors not have used higher rpm and consequently produce higher mass flows, than those produced? Otherwise, these experiments should still be considered under the lab-scale framework.

Figure 5 shows wide variations of the STD, this is mostly due to the narrow range used in the y-axis – the scales in the y-axis should be re-oriented to contain the full scale.

Author Response

Abstract

The final sentence in this section states that “The concept significantly reduces the material and instrumental costs of process design and implementation”. Can the authors provide a quantitative figure on this claim based on their study presented in this manuscript?

Answer: The calculation is not obvious. The experiments to determine the RTD can be made with only a few grams of API (1.5 g/impulse) and in a relatively short experiment design. The digital twin modeling of the feeders is non-destructive of the materials, so it can be more beneficial in the case of more expensive materials. After these low cost experiments, we were able to optimize the CPPs and the instrumentation, even using setups that would require investments, as in-silico experiments can be made where we used the same feeders for ASA and MCC feeding. These setups would require duplicates of these feeders to test in real life. The process control described in the study is using only the feeder data as soft sensor, therefore reducing the cost by a dedicated NIR spectrometer, as it is only needed for the RTD experiments. This has been showcased in the conclusions. (page 17, lines 494-498)

Introduction

Good overview of the various topics linked to continuous manufacturing, PAT, modelling but the rationale for selecting ASA as the API model system is not provided.

Answer: ASA was selected as the API model because this process is a part of a published end-to-end system with ASA as the API. This has been highlighted in the introduction. (page 3, lines 104-106)

Materials and Methods

- There are three paragraphs in the beginning of the section which are a copy/paste from the Journal’s guidelines and should not be there.

Answer: They have been deleted.

- Characteristics for the raw ASA and MCC used are missing (e.g. solid-state/polymorphic form for ASA, particle size and size distribution for ASA and MCC, etc.)?

Answer: The particle size distribution has been described in the materials section by the d10, d50 and d90 parameters of the ASA and MCC according to Malvern Mastersizer 2000 measurements. (page 3 lines 123-125)

Results

The pilot-scale experiments (Table 1) do not use a significantly higher rpm, should the authors not have used higher rpm and consequently produce higher mass flows, than those produced? Otherwise, these experiments should still be considered under the lab-scale framework.

Answer: The aim of the two experimental scale was to describe the effect of scale-up by changed mass flow on the system. However, as this study is connected to an end-to-end concept, we wanted to describe a design space that is applicable in this concept, in which the mass flow is limited by other production steps. This has been showcased in the introduction. (page 3 lines 106-109)

Figure 5 shows wide variations of the STD, this is mostly due to the narrow range used in the y-axis – the scales in the y-axis should be re-oriented to contain the full scale.

Answer: The figure has been changed to a uniform scale for MRT and sigma in the two one-at-a-time experimental designs.